# Contribution of Microbial Residues Obtained from Lignin and Cellulose on Humus Formation

**Shuai Wang** [1,2], **Nan Wang** [2], **Junping Xu** [2], **Xi Zhang** [2] **and Sen Dou** [1,*]

1   College of Resource and Environmental Science, Jilin Agricultural University, Changchun 130118, China
2   College of Agriculture, Jilin Agricultural Science and Technology University, Jilin 132101, China
*   Correspondence: dousen1959@126.com

**Abstract:** The contribution of microbial residues formed on lignin and cellulose to the formation of humus (HS) was investigated. The microbial residues formed by *Aspergillus niger* (*A. niger*) in the cultures of cellulose and lignin in a fluid medium were structurally characterized by elemental analysis, differential thermal analysis (DTA), FTIR spectroscopy and CP/MAS $^{13}$C NMR spectroscopy. Compared to cellulose itself, the microbial residue from cellulose contains more aromatic compounds and N-containing compounds and fewer carbohydrates and carboxylic compounds. *A. niger* improved the thermal stability and aromaticity of the cellulose. However, compared with that on lignin, more N-containing compounds, carbohydrates and carboxylic acid derivatives and less aromatic material were found in the microbial residue from lignin. Regardless of whether the carbon source was cellulose or lignin, *A. niger* utilized the N in the fluid medium to synthesize its own cells, and eventually, they could transfer the N into the microbial residue; in addition, the O-alkyl species dominated over the alkyl and aromatic compounds in the microbial residue. Although the molecular structures of the components of the microbial residue from lignin tended to be simpler, they were more alkylated, more hydrophobic and less aliphatic than those from cellulose. During culture with *A. niger*, the cellulose underwent degradation and then a polymerization, which led to an increased degree of condensation but a lower degree of oxidation, providing essential precursor substances for HSs formation. However, lignin underwent oxidative degradation. The microbial residue from lignin had a lower degree of condensation and a higher degree of oxidation.

**Keywords:** microbial residue; lignin; cellulose; *Aspergillus niger*; humus formation

## 1. Introduction

Accounting for 25% of the total organic C on the earth, humic substances (HSs) play an important role in terrestrial and aquatic systems, and they have bio-stimulatory effects on plant growth [1]. HSs are mainly generated via the polycondensation and polymerization of compounds formed during the decomposition of plant and microbial precursors, which results in HSs having compounds with unique chemical structures [2]. The microbial transformation of plant litter is a complex biochemical process and controls soil HSs formation [3]. As a substantial fraction of soil organic matter, the HSs are mostly derived from the decomposition of plant residues that consist mainly of cellulose and lignin [4]. The HSs in soil contain cellulose and lignin-derived molecules along with compounds derived from primary and secondary plant, animal and microbial metabolites [5].

Cellulose and lignin, which constitute the main skeletal components of plants and are widely found in various plant species, are the most abundant sources of C in the soil; cellulose is a linear polysaccharide, and lignin is a high-molecular-weight phenolic compound [6]. Lignin can be partly degraded and decomposed into smaller molecules or degraded to simple phenolic or aliphatic monomers [7], which are important precursors for HS formation. In addition, the degradation

of cellulose is easier and faster than that of lignin [4]. Therefore, lignin and cellulose contribute inconsistently to the formation of HSs.

Fungi are of great interest in the utilization of cellulose and lignin because they excrete some enzymes extracellularly [8]. The fungus Aspergillus niger is one of the most common species of the genus Aspergillus. It produces a number of enzymes, such as cellulases, xylanase, β-xylosidase, β-glucosidase and laccase, that can degrade cellulose and lignin [9]. Laccase is a phenol oxidase widespread in fungi including *A. niger*, and this enzyme is involved in the conversion of lignin into HSs in soil. Aspergillus spp. could grow efficiently on wheat straw, utilizing cellulose, lignin and some carbohydrates through the action of carbohydrases [10]. Chen et al. [9] investigated the degradation of rice straw, a type of recalcitrant lignocellulosic material, under mixed solid-state culture with Phanerochaete chrysosporium and *A. niger*. Barapatre and Jha [11] studied the biodegradation and bioconversion of lignin extracted under basic conditions by two potent brown-rot fungi (Aspergillus flavus and Emericella nidulans) and found that the fungi had degraded 19.0%~41.6% of alkali lignin (0.25%, w v$^{-1}$) within 21 days of incubation and reduced the colourity up to 14.4%~21.0%. *A. niger* could compensate for the deficiencies in β-glucosidase and cellobiohydrolase II from Trichoderma reesei and successfully improve cellulase production, enhancing the enzymatic hydrolysis of lignocelluloses [12]. The fungi could release organic acids to digest organic materials, and lower pH favours the growth of fungi, the breakdown of lignin or cellulose and the formation of HSs [13]. Therefore, it was thought that *A. niger* could play an important role in the degradation of cellulose and lignin, and their degradation products might be further modified through a variety of reactions, including polymerization, condensation and oxidation, to produce highly complex organic molecules that would facilitate the formation of HSs.

Microbes are the main driving force for the conversion of cellulose and lignin, and the microbial residues of these materials in soil are important parent materials for HS formation [14]. Although the positive effect of *A. niger* on the formation of HSs is undeniable, it is unknown whether the microbial residues formed by *A. niger* from cellulose and lignin have any effect on HS formation. A study of the structural characteristics of the microbial residues derived from cellulose and lignin should help clarify the heterogeneous structures of the HSs. Considering this issue, to elucidate the contribution of the microbial residues to the formation of HSs is very necessary. Liquid cultures in a shake flask were used to study the characteristic differences between the microbial residues derived from cellulose and lignin obtained from inoculation with *A. niger* at 15 days, 30 days and 75 days and 15 days, 45 days and 70 days, respectively. Elemental analysis, differential thermal analysis (DTA), Fourier transform infrared (FTIR) spectroscopy and 13C nuclear magnetic resonance (13C CP-MAS NMR) spectroscopy were applied to comprehensively explore the characteristic differences between the microbial residues derived from cellulose and lignin. The research on the lignin and cellulose humification pathway will provide more sustainable development and a utilization strategy for the utilization of agricultural resources, such as crop straw returning and organic material composting.

## 2. Materials and Methods

### 2.1. Preparations of A. niger *Inoculum and the Fluid Medium*

*A. niger* was isolated from black topsoil (0~20 cm in depth) that was collected from an experimental field under maize monoculture conditions at Jilin Agricultural University located in the area of 43°48′44″ N, 125°23′45″ E on May 15, 2018.

The purification and propagation of *A. niger* were performed on potato dextrose agar (PDA) in Petri dishes. The preparation process was as follows: *A. niger* was scraped off the Petri dishes, each dish was washed with 10 mL of sterile water, and then the inoculum was collected. The organic C content of the *A. niger* inoculum was 0.72 g L$^{-1}$.

Sodium carboxymethyl cellulose was purchased from Kemiou Chemical Reagent Co., Ltd. in Tianjin city, China. The molecular formula is RnOCH2COONa. Lignin (Cat No. 24104-32) was purchased from Japan Kanto Chemical Co., Ltd.

The recipe for the fluid medium was as follows: NaNO3 2.0 g, K2HPO4 1.0 g, KCl 0.5 g, MgSO4 0.5 g, FeSO4 0.01 g, and sodium carboxymethyl cellulose 20.0 g or lignin 0.5 g. The ingredients were diluted with sterile water to 1000 mL; the pH of the fluid medium did not need to be adjusted.

## 2.2. Experimental Design

Liquid cultures in shake flasks were prepared. One hundred millilitres of the fluid medium was weighed into a 250-mL Erlenmeyer flask, which was plugged with a germproof cotton plug, wrapped with a piece of newspaper, sterilized with high-pressure steam (at 121 °C, 20 min), and naturally cooled under sterile conditions. After cooling, the *A. niger* inoculum was added to the flask.

Four treatments were designed and included in this study: (1) L-An, which represents the microbial residue formed by *A. niger* in the fluid medium of lignin; (2) C-An, which represents the microbial residue formed by *A. niger* in the fluid medium of cellulose; (3) the original lignin sample, which is represented by CK-L; and (4) the original cellulose sample, which is represented by CK-C. The volumes of *A. niger* inoculum was set as 10 mL. But for the CK treatment, the inoculum was replaced by the sterile water of 10 mL. The liquid culture was agitated on a shaker at 180 r min$^{-1}$ at a constant temperature (28 °C). The Erlenmeyer flasks used for the cultures were weighed, and sterile water was added to the flasks at regular intervals to ensure that the total volume of fluid medium remained constant. Cellulose or lignin served as the C source, and the sampling times were set at 15 days, 30 days and 70 days and 15 days, 45 days and 70 days, respectively. At each sampling time, the supernatant and sediment were separated by high-speed centrifugation for 20 min (16,000 r min$^{-1}$, centrifugal force of 28,620 g). The sediment was transferred to an electrothermal blast drying oven for drying at 55 °C to a constant weight over 48 h, and then the solid was crushed and ground through a 0.01 mm sieve. The obtained powder was the microbial residue.

## 2.3. Analysis

The percentages of C, H and N in the microbial residue were measured by an elemental analyser (German Elementar Vario EL-C, H, N). The percentage of O + S was obtained by difference according to the following expression: % (O + S) = [100% − (%C + %H + %N)]. FTIR spectroscopy of the microbial residue was conducted with a Nicolet AV-360 infrared spectrometer (USA). DTA was conducted with a SHIMADZU TA-60 differential thermal analyser (Japan). 13C CP-MAS NMR spectra were acquired on a Bruker AV 400 NMR instrument (Switzerland) with a resonant frequency of 13C at 100.57 MHz, magic angle spinning frequency of 5 kHz, contact time of 2 ms, sampling time of 34 ms, circulatory delay time of 5 s and 2048 data points. The chemical shift was corrected relative to 2,2-dimethyl-2-silapentane-5-sulfonic acid sodium salt (DSS) as an external standard, and the peak area was calculated using the software package MestRe-C version 2.3a.

## 2.4. Statistical Analysis of the Data

Data analysis and spectrum processing were performed using Excel 2003 and Origin 8.0. The statistical analysis of the data in elemental composition, and heat, weight loss obtained from the DTA was conducted with SPSS 18.0 (ANOVA) with the least significant difference (LSD) and Duncan tests.

## 3. Results

### 3.1. Elemental Composition of Microbial Residue

The C/N, H/C and O/C atomic ratios were taken as indicators of the structure and the molecular shape of the microbial residue. The H/C ratios show the degree of maturity of the HSs since indirectly, it reflects the presence of more condensed aromatic rings or substituted ring structures. The O/C ratio is considered an indicator of the contents of carbohydrates and carboxylic acid derivatives [15]. As shown in Table 1, the C, H, O + S contents of C-An were all significantly lower than those of CK-C,

but C-An had higher N content. C-An showed lower C/N, H/C, and O/C atomic ratios than CK-C, indicating larger aromatic fractions (unsaturated structures), more N-containing compounds, and smaller contents of carbohydrates and carboxylic acid derivatives. The C content of C-An slightly decreased and then increased substantially during the culture period, while the N content of C-An gradually increased from 36.2 to 44.9 g kg$^{-1}$. These trends indicated that in the culture with *A. niger*, cellulose was first degraded, and then portions of the small-molecule degradation products could interact with the microbial cells to enter the residue together, resulting in the gradual increase in N-containing compounds.

**Table 1.** Elemental composition of the microbial residues formed by *A. niger* in the fluid medium of lignin and cellulose.

| Treatments | Culture Time (days) | Element Contents | | | | Atomic Ratios | | |
|---|---|---|---|---|---|---|---|---|
| | | C (g kg$^{-1}$) | H (g kg$^{-1}$) | N (g kg$^{-1}$) | O + S (g kg$^{-1}$) | C/N | H/C | O/C |
| C-An | 15 | 287.4 ± 3.5d | 48.9 ± 1.0c | 36.2 ± 0.8c | 425.2 ± 5.7b | 9.3 ± 0.6b | 2.0 ± 0.4b | 1.1 ± 0.1d |
| | 30 | 273.5 ± 2.8e | 46.9 ± 0.8e | 39.5 ± 0.9b | 402.6 ± 6.5c | 8.1 ± 0.8c | 2.1 ± 0.3b | 1.1 ± 0.2d |
| | 70 | 290.3 ± 4.5c | 47.9 ± 1.2d | 44.9 ± 1.1a | 383.9 ± 4.5d | 7.5 ± 0.6d | 2.0 ± 0.2b | 1.0 ± 0.1d |
| CK-C | - | 339.5 ± 6.4b | 61.6 ± 1.8b | 4.5 ± 0.3h | 594.5 ± 6.9a | 88.1 ± 0.5b | 2.2 ± 0.3ab | 1.3 ± 0.1c |
| L-An | 15 | 144.1 ± 1.4g | 26.8 ± 0.9g | 18.1 ± 0.9f | 323.3 ± 4.5f | 9.3 ± 0.2b | 2.2 ± 0.4ab | 1.7 ± 0.1b |
| | 45 | 145.0 ± 2.6f | 27.0 ± 0.7g | 20.7 ± 1.1e | 379.2 ± 3.9e | 8.2 ± 0.4c | 2.2 ± 0.2ab | 2.0 ± 0.2a |
| | 70 | 145.7 ± 1.9f | 27.8 ± 0.6f | 22.4 ± 0.7d | 389.5 ± 3.2d | 7.6 ± 0.3d | 2.3 ± 0.3a | 2.0 ± 0.1a |
| CK-L | - | 615.0 ± 8.6a | 68.9 ± 1.7a | 7.4 ± 0.4g | 308.6 ± 4.6g | 96.7 ± 1.8a | 1.3 ± 0.1c | 0.4 ± 0.0e |

C-An and L-An represent the microbial residues formed by *A. niger* in the fluid medium of cellulose and lignin, respectively; the original cellulose and lignin samples are represented by CK-C and CK-L, respectively. The same conventions are applied in the tables below. Values are mean±standard deviation from three replications. Based on the LSD and Duncan tests, different lowercase letters indicate a significant difference among the different treatments ($p < 0.05$), and a, b, c are ranking results.

The C and H contents in L-An were significantly lower than those of CK-L, but L-An had higher N and O + S contents. Compared with CK-L, L-An showed lower C/N and higher H/C and O/C ratios, which indicated more N-containing compounds, carbohydrates and carboxylic acid derivatives, and a smaller aromatic fraction. An increase in the H content indicated a greater proportion of aliphatic C moieties (CH2) than aromatic C moieties (C = C) [15]. During the culture period, the H content of L-An gradually increased from 26.8 to 27.8 g kg$^{-1}$, which also confirmed the occurrence of lignin degradation due to the increase in aliphatic C.

It is worth noting that with cellulose and lignin as the C sources (C-An and L-An), the C/N ratio followed the same trend, and both showed a gradual increases, while the H/C and O/C ratios were opposite; in C-An, these ratios decreased, while in L-An, these ratios increased. The microbial residues obtained using cellulose or lignin as the C source for *A. niger* could include different compounds.

### 3.2. DTA of Microbial Residue

Figure 1a shows the DTA curves of C-An. The DTA spectra from different culture durations were similar. There was a weak endothermic valley from 46 to 64 °C, and two obvious exothermic peaks at approximately 282 °C and from 475 to 483 °C. According to Table 2, the ratio of high heat release/medium heat release and the ratio of high weight loss/medium weight loss of the microbial residue gradually increased with culture time. As shown in Table 2, the DTA curve of the original sample of cellulose did not show an exothermic peak at a high temperature. However, after *A. niger* was added to the culture, an exothermic peak at a high temperature was observed.

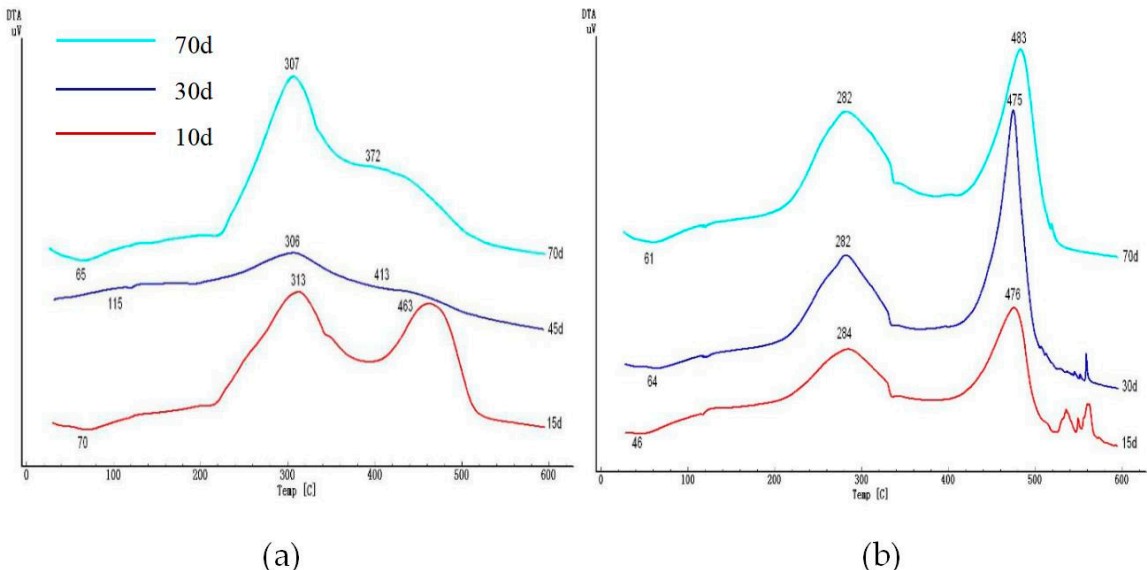

**Figure 1.** Differential thermal analysis (DTA) of the microbial residues formed by *A. niger* in the fluid medium of cellulose and lignin. (**a**) cellulose; (**b**) lignin.

**Table 2.** Parameters obtained from the differential thermal analysis (DTA) of the microbial residues formed by *A. niger* in the fluid medium of lignin and cellulose.

| Treatments | Culture Time (days) | Endothermic Peak | | Exothermic Peak I | | Exothermic Peak I | | Exothermic Peak II/ Exothermic Peak I | |
|---|---|---|---|---|---|---|---|---|---|
| | | Heat (kJ g$^{-1}$) | Weight Loss (%) | Heat (kJ g$^{-1}$) | Weight Loss (%) | Heat (kJ g$^{-1}$) | Weight Loss (%) | Heat | Weight Loss |
| C-An | 15 | 0.99 | 7.64 | 12.17 | 44.57 | 14.54 | 25.74 | 1.20c | 0.58b |
| | 30 | 0.59 | 7.94 | 11.0 | 40.56 | 17.62 | 28.35 | 1.60b | 0.70a |
| | 70 | 0.63 | 6.48 | 9.12 | 39.49 | 16.92 | 28.72 | 1.86a | 0.73a |
| CK-C | 0 | 0.34 | 4.86 | 9.73 | 40.97 | - | - | - | - |
| L-An | 15 | 0.64 | 6.15 | 12.00 | 42.86 | 9.00 | 23.69 | 0.75d | 0.55b |
| | 45 | 0.01 | 4.81 | 23.74 | 47.21 | 1.84 | 11.17 | 0.08f | 0.24d |
| | 70 | 0.95 | 6.84 | 11.60 | 37.10 | 1.99 | 16.45 | 0.17e | 0.44c |
| CK-L | - | 0.65 | 3.42 | 11.57 | 19.80 | - | - | - | - |

Based on the LSD and Duncan tests, different lowercase letters indicate a significant difference among the different treatments ($p < 0.05$), and a, b, c are ranking results.

Figure 1b shows the DTA curve of the microbial residue derived from lignin in the presence of *A. niger*. There was a weak endothermic valley within the range of 65 to 115 °C. In addition, two obvious exothermic peaks were observed at 306 to 313 °C and 372 to 463 °C, while the positions of the moderate and high-temperature exothermic peaks both gradually shifted to higher temperatures with increasing culture time. From the DTA data in Table 2, the ratio of high heat release/medium heat release and the ratio of high weight loss/medium weight loss from the microbial residues followed similar trends: They showed rapid initial decreases and then slow increases.

*3.3. FTIR Spectra of the Microbial Residue*

Combined with the absorption peaks in Figure 2 and the relative absorption peak intensities in Table 3, the FTIR spectra showed that the intensity of the absorption peak at 3414 cm$^{-1}$ (O–H stretching) in the spectrum of C-An decreased initially and then increased with culture time. The intensities of the two absorption peaks at 2925 cm$^{-1}$ and 2864 cm$^{-1}$ (aliphatic C–H stretching of CH3/CH2 groups) became weaker, and the absorption peak at 1721 cm$^{-1}$ (stretching vibration of C = O of carboxylic acid) was not obvious; its intensity gradually weakened. The vibration intensities of the two peaks

at 1644 cm$^{-1}$ (aromatic C = C skeletal vibrations) and 1412 cm$^{-1}$ (C–H asymmetric bending of CH3 groups) decreased. There was no obvious change in the intensity of the peak at 1326 cm$^{-1}$ (symmetric stretching vibration of carboxylates). The intensity of the peak at 1251 cm$^{-1}$ (phenol C–O stretching) gradually weakened, while the intensity of the absorption peak at 1061 cm$^{-1}$ (C–O stretching of polysaccharides) initially decreased and then increased. The (2925 + 2864)/1644 ratio first increased from 0.144 to 0.190 and then decreased to 0.113 at culture times of 15 d and 70 d.

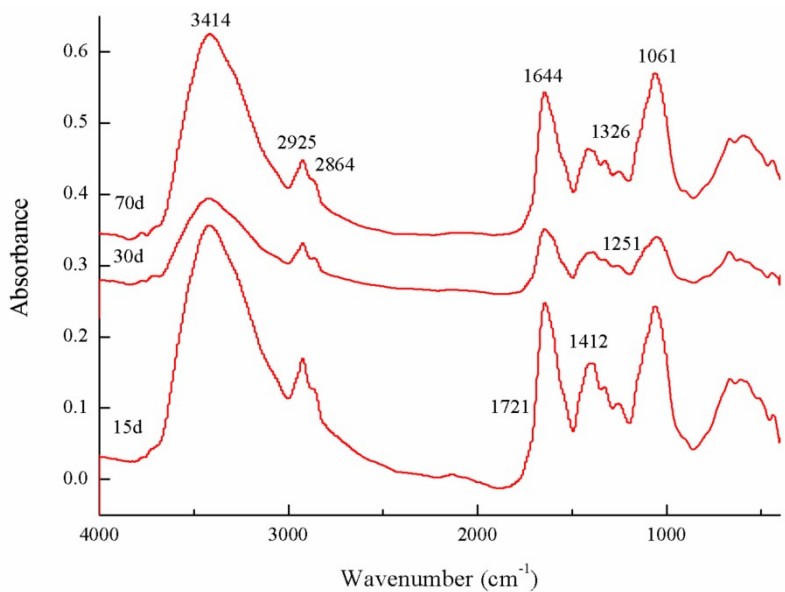

**Figure 2.** Fourier transform infrared (FTIR) spectra of the microbial residue formed by *A. niger* in the fluid medium of cellulose.

**Table 3.** Relative intensities of the main absorption peaks from the microbial residue formed by *A. niger* in the fluid medium of cellulose.

| Culture Time (days) \ Wavenumbers (cm$^{-1}$) | 3414 | 2925 | 2864 | 1721 | 1644 | 1412 | 1326 | 1251 | 1061 | (2925 + 2864)/1644 |
|---|---|---|---|---|---|---|---|---|---|---|
| 15 | 65.42 | 1.63 | 0.31 | 0.03 | 13.50 | 3.35 | 0.24 | 0.41 | 15.11 | 0.144 |
| 30 | 64.82 | 2.32 | 0.37 | 0.01 | 14.14 | 3.62 | 0.19 | 0.34 | 14.20 | 0.190 |
| 70 | 66.09 | 1.33 | 0.13 | 0.00 | 12.91 | 2.45 | 0.24 | 0.26 | 16.60 | 0.113 |

The FTIR spectra of L-An are shown in Figure 3. The spectra showed obvious absorption peaks at 3414 to 3344 cm$^{-1}$ (vibration of O-H from -OH), 2931 to 2873 cm$^{-1}$ (the vibration of C-H from aliphatic -CH3 or -CH2) and 1648 cm$^{-1}$ (aromatic C = C or C = O from benzene rings). The hydrogen bonds became stronger, the O-H bonds became longer, and the proportion of aliphatic compounds within the microbial residue increased with culture time. The (2931 + 2873)/1648 ratio of the microbial residue from lignin first increased and then decreased with culture time, and the ratio had an overall downward trend. Table 4 shows that there was no absorption peak at 1732 cm$^{-1}$ that could represent the stretching vibration of C = O moieties of carboxylic acids. The intensity of the peak at 1561 cm$^{-1}$ (N-H deformation or C-N stretching vibration) indicated that the proportion of N-containing compounds in the microbial residue first increased and then decreased in the culture of *A. niger*. During the culture period, the number of N-containing compounds increased, while the number of heterocyclic N compounds first decreased, then increased, and finally completely disappeared. The gradual increase in the intensity of the peak at 1385 cm$^{-1}$ (symmetric stretching vibration of carboxylates) showed that the content of symmetric organic carboxylic acids increased. The absorption peak at 1247 cm$^{-1}$ represented the stretching of C-O moieties in carboxyl groups (O-H deformation or C-O stretching vibration from aromatic ethers). The two peaks at 1142 cm$^{-1}$ and 1064 cm$^{-1}$ could represent

polysaccharides and carbohydrates, and the sum of their intensities increased from 15.15 to 21.41 with increasing culture time.

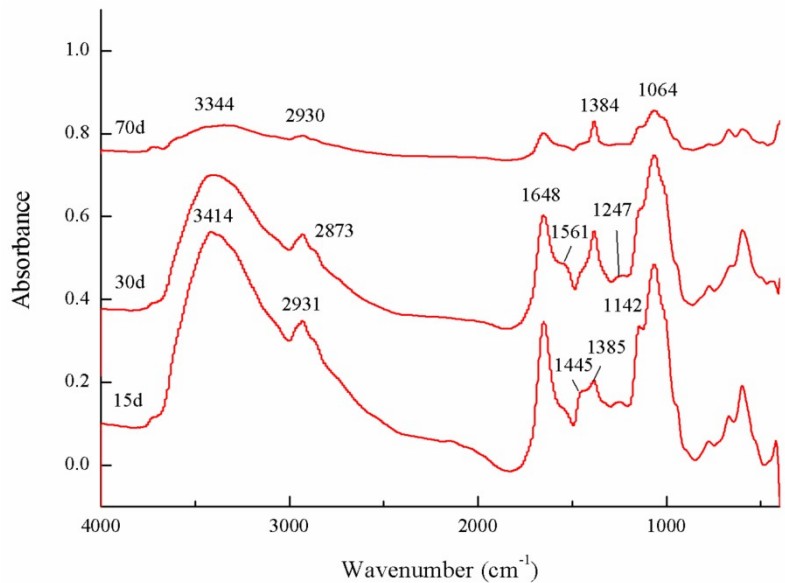

**Figure 3.** Fourier transform infrared (FTIR) spectra of the microbial residues formed by *A. niger* in the fluid medium of lignin.

**Table 4.** Relative intensities of the main absorption peaks from the microbial residue formed by *A. niger* in the fluid medium of lignin.

| Wavenumbers(cm$^{-1}$) Culture Time (days) | 3414 | 2931 | 2873 | 1732 | 1648 | 1561 | 1445 | 1385 | 1247 | 1142 | 1064 | (2931 + 2873)/1648 |
|---|---|---|---|---|---|---|---|---|---|---|---|---|
| 15 | 71.19 | 1.28 | 0.21 | 0.00 | 10.23 | 0.23 | 0.78 | 0.62 | 0.30 | 1.12 | 14.03 | 0.146 |
| 45 | 68.19 | 1.39 | 0.22 | 0.00 | 9.24 | 0.81 | 0.21 | 2.60 | 0.28 | 1.07 | 16.00 | 0.174 |
| 70 | 59.75 | 1.39 | 0.24 | 0.00 | 10.36 | 0.30 | 0.66 | 5.51 | 0.39 | 1.25 | 20.16 | 0.157 |

### 3.4. $^{13}$C CP-MAS NMR Spectra of Microbial Residue

The solid-state 13C NMR spectra of the microbial residues formed by *A. niger* in a fluid medium over lignin and cellulose are shown in Figure 4. The distributions of different C species were determined from the relative contributions of the signal intensities (Table 5). Both C-An and L-An exhibited weak peaks near 24 ppm due to methyl C (CH3) and methylene C (CH2)n [16]. According to the explanation by Joanisse et al. [17], the peak at 30 ppm mainly represents -CH2 groups in long chains derived from cutin, suberin and plant waxes. The largest peak in the O-alkyl region occurred at 74 ppm, and the sharp peak at 104 ppm was mainly due to carbohydrates. In the O-alkyl C region, there were some signals that indicated the presence of methoxy C at 55 to 61 ppm. The signal at 93 ppm in the spectrum of L-An corresponded to the anomeric C atoms of carbohydrates; however, the spectrum of C-An did not exhibit this peak. The weak peak was centred near 129 ppm, indicating carbons in a ring with a strong electron donor, such as O or N. The largest peak in the carbonyl-C region was at 175 ppm and was ascribed to the amide-C of proteins and the carboxyl groups of microbial lipids.

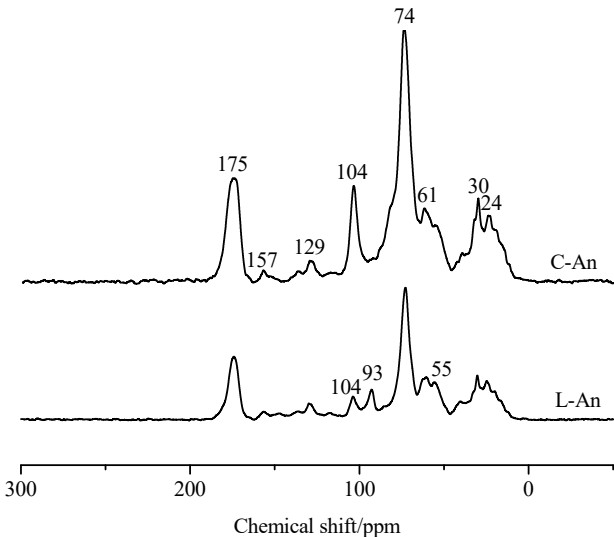

**Figure 4.** $^{13}$C nuclear magnetic resonance (NMR) spectra of the microbial residues formed by *A. niger* in the fluid medium of cellulose and lignin.

**Table 5.** Relative intensities (%) derived from the $^{13}$C nuclear magnetic resonance (NMR) spectra of microbial residues formed by *A. niger* in the fluid medium of lignin and cellulose at the final time.

| Treatments | 0–45 ppm | 45–110 ppm | 110–140 ppm | 140–160 ppm | 160–185 ppm | 185–220 ppm | Aromaticity Index (%) | Hydrophobic C/ Hydrophilic C | Alkyl-C/ O-alkyl-C |
|---|---|---|---|---|---|---|---|---|---|
| | Alkyl C | O-alkyl C | Aromatic C | Phenolic C | Carboxyl C | Carbonyl-C | | | |
| C-An | 19.66 | 60.84 | 4.93 | 0.99 | 13.9 | 0.67 | 5.70 | 0.40 | 0.32 |
| L-An | 21.77 | 55.17 | 8.73 | 2.77 | 14.3 | 0.04 | 9.87 | 0.55 | 0.39 |

Aromaticity index (%) = C ($\delta$110–160)/C($\delta$0–160) × 100%, Hydrophobic C/Hydrophilic C = [C($\delta$0–45) + C($\delta$110–160)]/[C($\delta$45–110) + C ($\delta$185–220)], Alkyl-C/O-alkyl-C = C($\delta$0–45)/C($\delta$45–110), and $\delta$ represents the chemical shift.

The relative areas were calculated as the percentage of the total intensity from the integral curves of the following C chemical-shift regions: alkyl C (0~45 ppm), O-alkyl C (45~110 ppm), methoxy C (50~60 ppm), aromatic C (110~140 ppm), phenolic C (140~160 ppm), carboxyl C (160~190 ppm) [18,19] and carbonyl C (190~220 ppm) [16,19]. Compared with L-An, C-An had larger proportions of O-alkyl C and carbonyl-C, and smaller proportions of alkyl C, aromatic C, phenolic C and carboxyl C. The aromaticity index (%) and hydrophobic C/hydrophilic C and alkyl-C/O-alkyl-C ratios of C-An were all less than those of L-An.

## 4. Discussion

Compared with that of cellulose, the degree of condensation of the microbial residue obtained from the culture of *A. niger* in the fluid medium of cellulose was higher, and the degree of oxidation was slightly lower. Meehnian et al. [20] indicated that certain fungi could utilize hemicellulose and decrease the H/C ratio in cotton stalks; thus, during the degradation of cellulose, more aromatic compounds tended to form [21]. However, compared with that from lignin, the microbial residue after the culture of *A. niger* had a lower degree of condensation, but the degree of oxidation was higher. Lignin has an affinity not only for carbohydrates through non-covalent bonds with aromatic compounds but also for hydroxy groups through covalent bonds with polysaccharides [22]. Therefore, it was clear that the polysaccharides produced by lignin degradation were easily adsorbed onto the surface of the microbial residue. The microbial residue derived from lignin underwent oxidative degradation during culture with *A. niger* in a fluid medium. Baddi et al. [23] thought that the levels of C and H decreased as composting progressed, while the level of O increased, and N remained more or less constant. The microbial residue derived from lignin in this study followed similar trends. Whether it was cellulose or lignin as the C source, *A. niger* could utilize the N in the culture

medium for cellular synthesis, and eventually, the cells could transfer N into the microbial residue. During the decomposition of agricultural litter, the N-containing compounds tend to accumulate [24]. Berg and Soderstrom [25] found an increase in fungal biomass during the decomposition of Scots pine needle litter that corresponded with an increase in the absolute amount of N in the needles. The accumulation of N-containing compounds appeared to be related to the tendency of N to become chemically immobilized during the humification process.

The decreased C/N and H/C ratios and the increased O/C ratio are the basis for the evaluation of compost maturity [23]. Compost generally contains lignin and cellulose components. According to the results of this study, it could be inferred that the cellulose components in the compost would lower the H/C ratio and that the lignin components would increase the O/C ratio.

There were few stable aromatic structures within the cellulose. *A. niger* enhanced the thermal stability and aromaticity of the microbial residue formed in the fluid medium of cellulose during culture. An exothermic peak at a high temperature, which could indicate a high degree of aromatization, was observed. Therefore, breaking the aromaticity requires a higher temperature, and the culture with *A. niger* favoured the transformation of cellulose into HSs with more complex structures. The culture with *A. niger* improved the thermostability of the microbial residue formed in the fluid medium of lignin. *A. niger* facilitated the formation of aliphatic compounds in the microbial residue in the early to middle stage of the culture (15~30 d). During this process, a larger proportion of aliphatic compounds were formed due to the degradation of lignin by *A. niger*. After that process, the condensation increased in the middle to late stage of the culture (30~70 days), while a large proportion of aliphatic compounds underwent microbial condensation reaction, resulting in an increase in the degree of aromaticity. The results of Bernabé et al. [26] suggested that at the beginning of the composting process, the degradation of lignin requires more heat and that the lignin extracted in the later period of the composting process showed more complex structures and a tendency toward humification, which could be attributed to the reactions that occur during composting, such as the beginning of the humification process and the formation of new stable species. However, at the end of the culture period, the contents of both stable materials and aromatic compounds in the microbial residue decreased, and their molecular structures tended to be simpler.

The inoculation of *A. niger* could cause the hydroxy content of the microbial residue to decrease initially and then increase, and the proportion of aliphatic compounds decreased. This effect might be because the rate of hydroxylation was lower than the rate of oxidation of cellulose during the first stage, while in the later stage, hydroxylation accounted for a large proportion of the processes within the culture [27]. The peaks at 2925 $cm^{-1}$ and 2864 $cm^{-1}$, associated with the C-H stretching of methyl or methylene groups, could indicate that the proportion of aliphatic compounds decreased, which implied that the methyl or methylene groups in cellulose were broken down. The stretching vibration of carboxyl C = O was not obvious and tended to decrease. The vibration intensity of aromatic C = C gradually decreased, and the methyl content decreased slightly. The deformation vibration of -OH derived from carboxyl groups and the intensity of the stretching vibration of C-O were weakened, while the stretching vibration of C-O from carbohydrates (or polysaccharides) initially became less intense and then become more intense with culture time. Therefore, the polysaccharides obtained from the degradation of cellulose by *A. niger* were the major carbon source and energy supply for the reproduction of *A. niger* in the early stage of the culture (15~30 days), after which the polysaccharide content further increased with the increasing degradation. Brink and Vries [28] reported that to enable the efficient degradation of cellulose, *A. niger* could produce a diverse set of carbohydrate-targeting enzymes. These enzymes have a carbohydrate-binding site that greatly enhances the efficiency of the degradation of cellulose microfibrils [29,30]. During the degradation of cellulose, *A. niger* tended to utilize polysaccharides to generate energy and better degrade cellulose, and then as the degradation progressed, and the substrates for *A. niger* were consumed, the consumption of polysaccharides decreased. The initial increase and then decrease in the (2925 + 2864)/1644 ratio indicated that the organic molecules in the microbial residue were first decomposed and then underwent condensation.

Finally, the structures of compounds in the microbial residue from cellulose tended to be complicated, resulting in the production of more aromatic C. In view of the strengthening of the hydrogen bond vibrations in the spectrum of the microbial residue formed from lignin, Crawford [31] reported that a large number of hydroxyl groups were generated during lignin biodegradation and transformation. Therefore, the hydroxylation degree was stronger [27]. In the report of Bi et al. [32], the degradation of lignin by Phoma herbarum was associated with the cleavage of β-O-4 linkages and afforded new phenolic hydroxy groups. Although *A. niger* enhanced the formation of aliphatic structures in the early to middle stages of culture, and in the later period, the degree of aromaticity increased, and the number of aliphatic compounds slightly decreased overall. Wei et al. [21] suggested that in the early stage of culture, the microbial degradation of lignin allowed more aliphatic C compounds to enter the microbial residue, and then the aliphatic C compounds were gradually condensed and transformed into aromatic compounds with more complex structures. The degradation of lignin by *A. niger* could produce a higher content of polysaccharides. Leisola et al. [33] studied the synthesis of polysaccharides by Phanerochaete chrysosporium during the degradation of kraft lignin, and the results showed that some fungi could produce various extracellular polysaccharides and that the excess polysaccharides formed inhibited the microbial degradation of lignin [34]. The polysaccharides produced by lignin degradation could be used by *A. niger* as an energy reserve, but the polysaccharides surrounded the *A. niger* cells, suppressing the absorption of more lignin, which might explain why the lignin was not completely degraded.

Regardless of if the C source was cellulose or lignin, O-alkyl C species dominated over alkyl C and aromatic C species in the microbial residue formed by *A. niger*. This was similar to the structure of humic acid reported by Nierop et al. [35] and humin reported by Zhang et al. [36]. The higher degree of aromaticity index and ratios of alkyl C/O-alkyl C and hydrophobic C/hydrophilic C generally indicated higher degrees of humification, alkylation, and hydrophobicity in the HSs [36]. Therefore, our results suggested that the microbial residue formed by *A. niger* in the fluid medium of lignin was more alkylated, more hydrophobic, and less aliphatic. This trend has also been observed in previous studies [19]. The microbial residue formed by *A. niger* in the fluid medium of cellulose contained no anomeric C signal, unlike the humic acid investigated by Al-Faiyz [16]. The microbial residue derived from lignin had a higher proportion of alkyl C groups, which was mostly due to the accumulation of recalcitrant structural components in the microbial residue derived from lignin, such as those of waxes, cutin, suberin, lipids, and amino acids. Meehnian et al. [20] showed that compared to untreated cotton stalks, pretreated cotton stalks became porous after lignin degradation and that hydrolytic enzymes were able to access the cellulose, which explained the strong hydrophobicity of lignin. The biodegradation of lignin was an oxidative process. In the bio-oxidation of lignin by Phanerochaete chrysosporium, the content of carboxyl and carbonyl groups increased, and that of methoxy and aliphatic hydroxy groups decreased [37,38].

Under the conditions of this experiment, the path of cellulose and lignin humification driven by *A. niger* is different, which may be one of the main causes of the structural heterogeneity of HSs molecules. Furthermore, straw returning and composting are important practices relevant to the humification of cellulose and lignin. The exploration of cellulose and lignin humification pathways will help to improve the sustainable utilization of agricultural waste.

## 5. Conclusions

*A. niger* used cellulose or lignin as the sole C source, and the contribution of humification to HSs formation was different. During culture with *A. niger*, the cellulose underwent degradation and then a polymerization, which led to an increased degree of condensation but a lower degree of oxidation, providing essential precursor substances for HSs formation. However, lignin underwent oxidative degradation. The microbial residue from lignin had a lower degree of condensation and a higher degree of oxidation.

**Author Contributions:** "conceptualization, S.W. and S.D.; methodology, S.D.; software, J.X.; validation, S.W., N.W. and X.Z.; formal analysis, S.D.; investigation, S.W.; resources, S.D.; data curation, S.W.; writing—original draft preparation, S.W.; writing—review and editing, S.W.; visualization, S.W.; supervision, S.D.; project administration, S.W.; funding acquisition, S.D."

**Funding:** This research was financially supported by the National Natural Science Foundation of China (Grant Nos. 41401251 and 41571231).

**Conflicts of Interest:** The authors declare no conflict of interest.

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
