# Peer review of "Contribution of Microbial Residues Obtained from Lignin and Cellulose on Humus Formation"

_sustainability, doi:10.3390/su11174777_

Round 1
Reviewer 1 Report
The manuscript is presented on contribution of microbial residues obtained from lignin and cellulose on humus formation. A significant effort has been taken in this research with interesting results and characterization studies. Some minor comments as follows need to be addressed:
1-L9: ‘the’ need to be replaced by ‘The’.
2-L22-24: This statement need revision to be clear to the reader: ‘During culture with A. niger, the cellulose underwent degradation and re-growth, which led to an increased degree of condensation but a lower degree of oxidation, providing essential precursor substances for HSs formation.’
3-L102: Why such a low volume (10 ml) was considered for testing in 250 ml flask?
4-Section 2.4: What kind of experimental design was used for statistical analysis?
5-Table 1: What type of multiple range test used for ranking? Please indicate underneath the Table. Also, indicate a,b,c… are ranking results.
6-Table 2. Please make a correction on the last column and the units of heat and weight loss.
7-Discussion is presented very well, however, conclusion section should be presented in a separate section.
Reviewer 2 Report
This paper concerns the breakdown of cellulose and lignin by Aspergillus niger in a laboratory simulation of leaf litter and humus metabolism.
There are some issues that should be addressed to improve the paper:
The introduction sets the scene but is somewhat repetitive. It could be tightened, and the wider context of sustainability which is relevant to the study should be further explored.
The discussion does not return to the wider context. The discussion should refer back to issues raised in the introduction. Also, since this has been submitted to Sustainability, there should be much more written on how the study relates to matters of sustainability. I could see very little on this.
The paper would benefit from a summary and conclusion.
Specific issues:
Discussion: The paragraphs are very long and need splitting up. A paragraph should only have one thought in it.
Line 50 and many others: In my opinion, it is best just to state the finding or insight and then add the reference. That is, do not name the authors in the text directly but concentrate on the information. For example, "Aspergillus spp. could grow efficiently on wheat straw, 50 utilizing cellulose, lignin and some carbohydrates through the action of carbohydrases [10]." This makes the text much simpler to read while retaining the reference citation appropriately.
Line 70, "Considering this issue..." is not a full sentence.
Line 71, "Liquid cultures..." belongs in the methods (What did we do?) section.
Line 172. Please spell out DA as the abbreviation has not been defined, nor is it used again. Abbreviations should be used, it at all, only when the concept is used frequently in a paper.
Figure 2 and others. In table and figure titles you should spell out in full any abbreviations. The table and figure should be understandable without recourse to the text.
Figure 2. Do you mean "wavelength" for "wave number"?
Tables 3 & 4. Why not use the full ratio in the table? Why use a footnote?
Table 5 needs reformatting.
Reviewer 3 Report
The author investigates the effect of A. niger on the microbial residue formed from lignin and cellulose. By using elemental analysis, DTA, FTIR and NMR, the author characterizes the chemical composition of above samples. The author argues that the thermal stability and aromaticity of cellulose have been improved by A. niger, and cellulose has more condensation but lower degree of oxidation. However, more N-containing components and less aromatic material are formed in microbial residue of lignin. In addition, microbial residue of lignin has lower condensation and higher oxidation.
Though the experiments are solid, there are still a few details in the manuscript that need to be improved:
In figure 1, figure 2 and figure 3, there is no scale and ticks on y axis, which is confusing. In figure 1, it would be better to add a visible legend to show which color stands for which experimental condition. Besides, the labels of x and y axis are too small for visualization purpose.
Round 2
Reviewer 2 Report
The authors of this R1 version of their paper on the contribution of Aspergillus species to the formation of humus substances from lignin and cellulose have answered the points raised by the first review.
The paper has merit. However, I am not convinced that this paper has enough on sustainability, either in the introduction or the discussion, to interest readers of Sustainability. The paper is heavy on chemistry but limited in scope on sustainability issues (see https://www.mdpi.com/journal/sustainability/about for details on the aims and contents of the journal). There is nothing in the abstract about sustainability issues.
The authors need to consider the aims of the journal to which they are submitting the paper and write their paper accordingly.
The authors might find help in finding a suitable journal from the MDPI list of journals on https://www.mdpi.com/about/journals or using JANE (Journal/Author Name Estimator) at http://jane.biosemantics.org/